# Unraveling the In Vitro Toxicity Profile of Psychedelic 2C Phenethylamines and Their *N*-Benzylphenethylamine (NBOMe) Analogues

**DOI:** 10.3390/ph16081158

**Published:** 2023-08-15

**Authors:** Daniel Martins, Eva Gil-Martins, Fernando Cagide, Catarina da Fonseca, Sofia Benfeito, Carlos Fernandes, Daniel Chavarria, Fernando Remião, Renata Silva, Fernanda Borges

**Affiliations:** 1CIQUP-IMS/Department of Chemistry and Biochemistry, Faculty of Sciences, University of Porto, Rua do Campo Alegre s/n, 4169-007 Porto, Portugalcarlos.fernandes@fc.up.pt (C.F.); daniel.chavarria@fc.up.pt (D.C.); 2Associate Laboratory i4HB—Institute for Health and Bioeconomy, Faculty of Pharmacy, University of Porto, 4050-313 Porto, Portugal; up201804901@edu.ff.up.pt (C.d.F.); remiao@ff.up.pt (F.R.); 3UCIBIO—Applied Molecular Biosciences Unit, Laboratory of Toxicology, Department of Biological Sciences, Faculty of Pharmacy, University of Porto, 4050-313 Porto, Portugal

**Keywords:** new psychoactive substances, 2C drugs, NBOMe drugs, SH-SY5Y cells, HepG2 cells, neurotoxicity

## Abstract

Mescaline derivative (2C phenethylamines) drugs have been modified by the introduction of a *N*-2-methoxybenzyl group to originate a new series of compounds with recognized and potent psychedelic effects, the NBOMe-drugs. Although they are prevalent in unregulated drug markets, their toxicity profile is still poorly understood, despite several reports highlighting cases of acute intoxication, with brain and liver toxicity. Thus, in this study, mescaline, 2C-N (insertion of a nitro in the *para* position of the 2C phenethylamines aromatic ring) and 2C-B (insertion of a bromide in the *para* position of the 2C phenethylamines aromatic ring) and their corresponding NBOMe counterparts, mescaline-NBOMe, 25N-NBOMe and 25B-NBOMe, were synthetized and the in vitro neuro- and hepatocytotoxicity evaluated in differentiated SH-SY5Y and HepG2 cell lines, respectively. Cytotoxicity, oxidative stress, metabolic and energetic studies were performed to evaluate the main pathways involved in their toxicity. Our results demonstrated that the presence of the *N*-2-methoxybenzyl group significantly increased the in vitro cytotoxicity of 2C phenethylamines drugs in both cell lines, with the NBOMe drugs presenting lower EC_50_ values when compared to their counterparts. Consistently, our data showed a correlation between the drug’s lipophilicity and the EC_50_ values, except for 2C-B. The 2C-B presented higher cytotoxic effects in both cell lines than mescaline-NBOMe, a result that can be explained by its higher passive permeability. All the NBOMe derivatives were able to cross the blood–brain barrier. Considering metabolic studies, the cytotoxicity of these drugs was shown to be influenced by inhibition of cytochrome P450 (CYP), which suggests a potential role of this enzyme complex, especially CYP3A4 and CYP2D6 isoenzymes in SH-SY5Y cells, in their detoxification or bioactivation. Furthermore, in differentiated SH-SY5Y cells, the drugs were able to induce mitochondrial membrane depolarization, and to disrupt GSH and ATP intracellular levels, these effects being concentration dependent and more pronounced for the NBOMe derivatives. No ROS overproduction was detected for any of the drugs in the tested experimental conditions. A correlation between a drug’s lipophilicity and the EC_50_ values in both cell lines, except for 2C-B, was also obtained. In summary, the introduction of a NBOMe moiety to the parent drugs significantly increases their lipophilicity, brain permeability and cytotoxic effects, with GSH and ATP homeostasis disruption. The inhibition of CYP3A4 and CYP2D6 emphasized that CYP-mediated metabolism impacts the toxicity of these drugs.

## 1. Introduction

The use of psychedelic drugs for religious, medicinal and recreational purposes has a long history in human civilizations [1]. These compounds are usually represented by classic serotonergic psychedelics—which act as agonists or partial agonists of serotonin (5-HT) receptors—and include psilocybin (present in “magic mushrooms”), *N*,*N*-dimethyltryptamine (DMT), mescaline and lysergic acid diethylamide (LSD) [1,2] (Figure 1). In general, these drugs are considered physiologically safe, presenting low potential for addiction or compulsive-seeking behaviors [1,2]. Reports of psilocybin, DMT, mescaline or LSD deaths resulting from overdose are unknown, including with high doses [3,4,5,6]. However, fatal accidents related to dangerous behavior after use have been reported [7,8]. Despite the safety profile and therapeutic potential of classic psychedelics, they were included in Schedule I of the Controlled Substances Act of 1970, mostly due to their growing popularity as recreational drugs. The search for legal alternatives to these drugs led to the emergence of new psychoactive substances (NPS) on the global drug market [9,10].

The number and diversity of NPS able to circumvent international drug laws dramatically increased in the last decades. Between 2009 and 2021, 134 countries and territories worldwide identified and reported 1124 substances to the United Nations Office on Drugs and Crime (UNODC). Up to December 2020, the European Monitoring Centre for Drugs and Drug Addiction (EMCDDA) was tracking around 830 NPS, mainly from the synthetic drug subclasses cathinones, cannabinoids and phenethylamines [11].

Phenethylamines are among the largest NPS groups introduced in unregulated markets to mimic *classic* illicit drugs [11,12]. The 2,5-dimethoxyphenethylamine-based drugs (2C phenethylamines drugs), gained popularity after publication of “PIHKAL” [13], a compendium where the synthesis, dosages and effects of more than 200 psychoactive compounds were reported. More recently, potent analogues of the 2C phenethylamine drugs, the well-known *N*-(2-methoxybenzyl)phenethylamines (NBOMes), were reported [14].

The effects induced by 2C phenethylamines and NBOMes drugs are mainly mediated by the activation of the 5-HT_2A_ receptor. The presence of a NBOMe substituent significantly increases the drugs affinity for the 5-HT_2A_ receptor, consequently increasing their potency [15,16]. While the subjective effects of NBOMes reported by users resemble classic psychedelics, the consumption of high doses was correlated with severe intoxications and deaths. Serious clinical conditions reported include seizures, metabolic acidosis, rhabdomyolysis, renal failure, multi-organ failure and coma [17,18,19,20]. Currently, it is not clear whether the deaths resulted from overdose or from the inherent toxicity of the compounds [1,15]. This uncertainty is mainly due to the lack of precise toxicological information.

In the present study, *β*-phenethylamines (mescaline, 4-bromo-2,5-dimethoxyphenethylamine (2C-B), 2,5-dimethoxy-4-nitrophenethylamine (2C-N)) and their corresponding *N*-benzyl-*β*-phenethylamines (Figure 2) were synthesized. To address the toxicological information gap, we evaluated their cytotoxic profile, assessing some putative mechanistic pathways involved in their cytotoxicity at brain and liver target organs, using differentiated SH-SY5Y and HepG2 cells, respectively. The drug-like properties of the *β*-phenethylamines and *N*-benzyl-*β*-phenethylamines in analysis, namely lipophilicity, interaction with phospholipidic membranes and permeability, were evaluated and correlated with the in vitro cellular effects, namely at oxidative stress, metabolic and energetic levels.

## 2. Results and Discussion

### 2.1. Chemistry

The synthetic strategies followed for the obtention of *β*-nitrostyrene, *β*-phenethylamines and *N*-benzyl-*β*-phenethylamines were adapted from the literature [13,21,22]. Briefly, the *β*-nitrostyrene derivatives were synthesized by a microwave-assisted Henry reaction. The 2C phenethylamines drugs were synthesized by the reduction of the respective *β*-nitrostyrene and 4-bromo-2,5-dimethoxy-β-phenethylamine (2C-B) and 5-dimethoxy-4-nitro-dimethoxy-β-phenethylamine (2C-N) were synthesized from 2,5-dimethoxy-*β*-phenethylamine. The *N*-benzylphenethylamine (NBOMe) derivatives were obtained from the corresponding *β*-phenethylamines by an indirect reductive amination process.

### 2.2. Evaluation of Drug Lipophilicity and Interaction with Phospholipidic Membranes

The ability of small molecules to pass through biological membranes influences their concentration on biologically relevant sites, therefore influencing their biological and/or toxic outcomes [23,24]. Passive diffusion is the main mechanism involved in the crossing of small molecules through biological membranes [23,24]. The lipophilicity and the affinity of small molecules to interact with phospholipidic membranes are two of the key parameters that influence their passive permeability [24,25].

The measurement of lipophilicity through the determination of Chromatographic Hydrophobicity Index (CHI) by fast analytical HPLC measurements relies on the partition of an HPLC stationary reverse phase into an aqueous/organic mobile phase and the Immobilized Artificial Membrane (IAM) is a biomimetic HPLC technique based on the use of stationary phase that mimics the lipid environment found in cell membranes [23,24,25].

Hence, to better understand the influence of these two parameters on the toxicity profile of the drugs under study, the lipophilicity (CHILogD_7.4_) and the passive permeability (K_pcell_) were back calculated from the respective CHI and CHI(IAM) values [)] (Table 1, Appendix A). In our experimental conditions it was not possible to evaluate CHI and CHI IAM for mescaline hydrochloride (Mescaline.HCl).

From the obtained data it is possible to observe that, as expected, the insertion of a bromide in the *para* position of the aromatic ring (2C-B) increased the lipophilicity (measured by CHILogD_7.4_) in comparison with the 2C-N. Furthermore, the insertion of the 2-methoxybenzyl group led to a higher lipophilicity when compared with their parent drugs (25B-NBOMe vs. 2C-B and 25N-NBOMe vs. 2C-N). From the studied drugs, 25B-NBOMe was the one with the highest lipophilicity. The same interpretation of the results can be taken from the Kpcell data (Appendix A). However, it was possible to observe that 2C-B has a higher passive permeability than mescaline-NBOMe despite presenting a lower lipophilicity (Table 1). Hence, to expand our knowledge, other parameters were assessed that, together with lipophilicity, are known to impact the passive permeability of small molecules. These parameters include the amount of hydrogen bond donors (HBD) and the number of rotatable bonds (NRB) and topological polar surface area (tPSA) (Table 1). The number of hydrogen bond acceptors were retained throughout the drugs evaluated in this work.

Although all the drugs have the same number of HBD, the addition of the 2-methoxyphenyl group led to a small decrease on the TPSA, suggesting a small impact of these parameters on the differences observed between the cellular permeability. However, when comparing the number of rotational bonds of the drugs it was possible to observe that, whilst the increase of this parameter seems to be compensated by the higher lipophilicity of 25B-NBOMe and 25N-NBOMe, in mescaline-NBOMe we observed a lower cellular passive permeability than 2C-B, despite presenting a higher lipophilicity. According to SwissADME, the drugs that were predicted to have the ability to cross the blood–brain barrier (BBB) were only the derivatives based on NBOMe (Table 1).

### 2.3. Evaluation of Drugs BBB Permeability

Passive permeability through the BBB is a key parameter that influences the concentration of small molecules capable of accumulating in the brain tissues and, therefore, the extent to which they can promote a biologic/toxic effect [26].

In the present study we used a Parallel Artificial Membrane Permeation Assay (PAMPA) with a membrane mimicking the BBB phospholipidic composition (PAMPA-BBB) to evaluate and predict passive, transcellular permeability of the mescaline-NBOMe, 25B-NBOMe and 25N-NBOMe drugs. The results are described in Table 2.

The good correlation (Appendix A) attained (log Pe (exp) = 0.9013 log Pe (rep.) − 1.059, R^2^ = 0.9690) allowed us to establish the ranges of permeability as (a) ${\mathrm{CNS}}^{+}$: −log Pe < 5.0 for compounds with high permeability (i.e., can enter the CNS); (b) ${\mathrm{CNS}}^{-}$: −log Pe > 6.8, for compounds with low permeability (i.e., excluded from the CNS); and ${\mathrm{CNS}}^{\pm }$: 5.0 < −log Pe < 6.8, for compounds with uncertain permeability.

The PAMPA-BBB data are in accordance with the computationally predicted values, confirming that MESC-NBOMe, 25B-NBOMe, 25N-NBOMe can successfully cross the BBB by passive diffusion. A correlation was observed between the lipophilicity of the drugs and their passive permeability through the BBB. Together, the observed results suggested that lipophilicity may have a direct impact on the permeability of the NBOMe drugs through the BBB.

### 2.4. Neurotoxic Profile of 2C Phenethylamines and Their NBOMe Counterparts

As the brain is a target organ to this type of drug exposure, their cytotoxicity (after 24 h exposure) was evaluated in differentiated SH-SY5Y cells by the neutral red uptake and the resazurin reduction assays. Figure 3 illustrates the obtained concentration–response (cell death) curves and in Table 3 are depicted the parameters of the fitted curves, namely, the baseline (bottom), the maximum cell death (top), the Hill Slope and the half-maximum-effect concentration (EC_50_). In Table 3 is also illustrated the curve *p* value obtained for the overall comparison between the 2C-X and 25X-NBOMe curves. The EC_50_ values of the fitted curves were used for statistical comparison of drug-induced cytotoxicity.

As observed in Figure 3, the 2C phenethylamines and their NBOMe counterparts under study caused a significant and concentration-dependent cytotoxic effect on SH-SY5Y cells by both used methodologies, except for mescaline, where no significant cytotoxic effects were detected by the resazurin reduction assay. Additionally, the NBOMe drugs show more remarkable cytotoxic effects than the corresponding 2C drugs, resulting in a noteworthy and significant shift of the NBOMe curves to the left (Figure 3), when compared to the 2C curves. Consequently, NBOMe drugs present significantly lower EC_50_ than 2C drugs (Table 3).

Mescaline showed no significant cytotoxic effects in the resazurin assay and the lowest cytotoxicity observed in the neutral red uptake assay, considering the tested conditions (Figure 3). Thus, for this compound, it was not possible to obtain a complete concentration–response curve in the range of concentrations tested. However, the presence of a 2-methoxybenzyl substituent—mescaline-NBOMe—significantly increased its cytotoxicity, as observed in Figure 3, with mescaline-NBOMe showing an EC_50_ value of 405.6 µM in the neutral red uptake assay and of 677.2 µM in the resazurin reduction assay (Table 3). On the other hand, 25B-NBOMe was the most cytotoxic drug, showing, among the tested drugs, the lowest EC_50_ values for both cytotoxicity assays (33.86 µM in the neutral red uptake assay and 58.36 µM in the resazurin reduction assay) (Table 3).

Overall, and as previously mentioned, the drug-induced cytotoxicity was significantly increased by the presence of a 2-methoxybenzyl group on the amine function of the 2C phenethylamines, with the NBOMe drugs showing lower EC_50_ values when compared to their counterparts (Table 3). Moreover, in the present experimental conditions, the neutral red uptake cell viability assay was revealed to be more sensitive than the resazurin reduction assay, as demonstrated by the lower EC_50_ values obtained for all drugs (Table 3). Therefore, this methodology was selected to be used in the subsequent experiments.

### 2.5. Effect on Neuronal Oxidative Stress

Reactive oxygen species (ROS) production was evaluated using the 2′,7′-dichlorofluorescein diacetate (DCFH-DA) probe. In the cytoplasm, DCFH-DA is hydrolyzed producing 2′,7′-dichlorodihydrofluorescein (DCFH) that, in the presence of ROS, is easily oxidized to the highly fluorescent 2′,7′-dichlorofluorescein (DCF) [27].

According to the obtained results, none of the drugs under study, for any of the concentrations tested and 24 h after exposure, caused a significant change in the ROS intracellular levels (Appendix A). Moreover, *t*-BHP (200 µM), used as positive control, caused, as expected, a significant increase in ROS intracellular levels, when compared to negative control cells. The data showed that the chemical functions present in the drugs do not take part in the redox reactions related with the amplification of ROS production.

### 2.6. Effect of Cytochrome P450 Inhibition on Drug-Induced Neurotoxicity

Differentiated SH-SY5Y cells were incubated with the drugs in the presence or absence of different CYP inhibitors: 1000 μM ABT, a non-selective CYP inhibitor, 10 μM quinidine, a specific CYP2D6 inhibitor, or 10 μM ketoconazole, a specific CYP3A4 inhibitor.

As observed in Figure 4, in ABT-treated cells, a significant decrease in neutral red uptake, demonstrative of an increased cytotoxicity, was observed for the compound’s mescaline-NBOMe (500 µM) and 25N-NBOMe (100 µM), while a significant increase in neutral red uptake (demonstrating a decreased cytotoxicity) was observed for 2C-N (50 µM) upon a non-selective CYP inhibition. Regarding CYP2D6, we observed that the inhibition of CYP2D6 by quinidine had a significant impact on the cytotoxicity of all drugs. It notably resulted in a significant decrease in the neutral red uptake for the following compounds: mescaline (500 µM), mescaline-NBOMe (250 and 500 µM), 2C-B (250 µM), 25B-NBOMe (50 µM), 2C-N (50, 100, 250, and 500 µM) and 25N-NBOMe (50 and 100 µM). These findings demonstrate an increased cytotoxicity when CYP2D6 is inhibited, thus highlighting the role of CYP2D6 in the detoxification of the drugs being studied. Similarly, the ketoconazole-mediated CYP3A4 inhibition resulted in a small but significant decrease in neutral red uptake (increased cytotoxicity) for mescaline (250 and 500 µM), mescaline-NBOMe (250 and 500 µM), 2C-B (100 and 250 µM) and 25N-NBOMe (50 and 100 µM), suggesting the involvement of CYP3A4 in the detoxification of such drugs.

These results suggest that the cytochrome P450-mediated metabolism has an impact on the cytotoxicity of these drugs, generally functioning as a detoxification pathway, since its inhibition results in higher cytotoxic effects. The exception was the drug 2C-N (50 µM) where ABT-mediated non-selective CYP inhibition resulted in a significant decrease in 2C-N cytotoxicity, suggesting a CYP-mediated bioactivation pathway, although involving an isoenzyme distinct from CYP2D6 and CYP3A4.

### 2.7. Effect of Monoamine Oxidase Inhibition on Drug-Induced Neurotoxicity

Monoamine oxidases (MAOs) are responsible for catalyzing the oxidative deamination of monoamines, including monoamine neurotransmitters. In humans, MAO is present in two isoforms—MAO-A and MAO-B [28]. Therefore, the inhibition of MAO-A and MAO-B was used to explore the impact of MAO-mediated metabolism on the cytotoxicity of these drugs.

Accordingly, differentiated SH-SY5Y cells were co-incubated with the drugs in the presence or absence of two MAO inhibitors, clorgyline (a MAO-A inhibitor) or rasagiline (a MAO-B inhibitor). As portrayed in Figure 5, clorgyline-mediated MAO-A inhibition significantly increased the cytotoxicity of mescaline (50, 100 and 250 µM), mescaline-NBOMe (100 and 250 µM) and 25N-NBOMe (50 and 100 µM), as observed by the significant decrease in the neutral red uptake, while rasagiline-mediated MAO-B inhibition significantly increased the cytotoxicity of mescaline-NBOMe (500 µM), 2C-B (100 and 250 µM) and 2C-N (500 µM), as observed by the significant decrease in the cell viability. Overall, the obtained results highlight that the MAO-mediated metabolism can operate as a potential detoxification pathway, significantly decreasing drug-induced cytotoxicity. On the other hand, for some drugs, clorgyline- and rasagiline-mediated MAO inhibition resulted in a significant increase in cell viability, as observed by the significant increase in neutral red uptake obtained for mescaline-NBOMe (with 1 µM clorgyline for the 500 µM drug concentration), 25B-NBOMe (with 1 µM rasagiline for the 50 and 100 µM drug concentrations) and 25N-NBOMe (with 1 µM rasagiline for the 50 and 100 µM drug concentrations). The data put forward the hypothesis that, for these specific cases, the MAO-mediated metabolism can be a potential bioactivation pathway (as the inhibition of MAO-mediated drug metabolism resulted in a decreased drug-induced cytotoxicity).

The effects of the drugs under study on MAO activity were then evaluated in a cell-free system, using human recombinant MAO-A and MAO-B (*h*MAO-A and *h*MAO-B, respectively) and kynuramine, a non-selective MAO substrate. The results presented in Appendix A showed that the drugs at 10 µM did not present significant *h*MAO-A inhibition properties (percentages of inhibition ranged between 3.9% and 17.5%). However, although most compounds were also poor *h*MAO-B inhibitors, the percentages of inhibition towards *h*MAO-B were higher than those obtained towards *h*MAO-A (18.0–89.0%). Mescaline-NBOME was the only compound with significant *h*MAO-B inhibition properties, displaying a percentage of *h*MAO-B inhibition of 89% at 10 µM.

In summary, the drugs under study have modest effects on MAO-mediated metabolism. At high concentrations (high µM range), the toxicity associated with the interference in MAO activity is not completely excluded. However, the toxic effects of these compounds at lower concentrations may result from a more prominent interference in other cellular processes.

### 2.8. Effect of the Drugs on the Neuronal Mitochondrial Membrane Potential

Mitochondrial transmembrane potential (Δѱm) was determined using the JC-1 dye, which can selectively enter the mitochondria and reversibly change the dye’s fluorescent properties depending on the Δѱm [29]. At high Δѱm conditions—representative of healthy cells—JC-1 forms J-aggregates generating red fluorescence. At low Δѱm conditions—representative of unhealthy or apoptotic cells—JC-1 is predominantly a monomer generating green fluorescence. Thus, a reduction in the red/green fluorescence ratio reveals mitochondrial membrane depolarization, while an increase reveals mitochondrial membrane hyperpolarization [30].

As shown in Figure 6, a significant and concentration-dependent decrease in the JC-1 red/green ratio was observed for all the studied drugs in differentiated SH-SY5Y cells. Some of the tested compounds induce a noticeable decrease in the JC-1 red/green ratio when compared to CCCP, a potent mitochondrial uncoupling agent, thus revealing the potential of the tested drugs to invoke the loss of mitochondrial membrane integrity by promoting mitochondrial membrane depolarization. Although all drugs showed a significant reduction in the JC-1 ratio, this effect was more evident for the NBOMe drugs, with 25B-NBOMe being the most operative drug in inducing mitochondrial membrane depolarization. These results are thus in agreement with the obtained cytotoxicity data, where 25B-NBOMe was identified as the most cytotoxic drug.

### 2.9. Effect of the Drugs on Intracellular Glutathione Levels

Total GSH was quantified through the DTNB-GSH recycling assay. Briefly, GSH is oxidized by 5,5′-dithiobis (2-nitrobenzoic acid) (DTNB) producing 5-thio-2-nitrobenzoic acid (TNB) and a glutathione adduct of GSH (GS-TNB). In the presence of NADPH, GS-TNB is reduced by glutathione reductase to 2GSH (recycled). The GSH concentration in the samples is thus proportional to TNB formation, which is measurable at 415 nm [31].

The intracellular levels of total GSH were evaluated 24 h after exposure of differentiated SH-SY5Y cells to the tested drugs. As depicted in Figure 7, a significant and concentration-dependent decrease in total GSH intracellular levels was observed for all tested drugs. Moreover, this decrease was more pronounced for the NBOMe drugs when compared to their 2C counterparts. For example, no significant alterations in the total GSH intracellular levels were observed for 2C-B when tested at the 250 μM concentration, while 25B-NBOMe, tested at a 10 μM concentration (25 times less concentrated), drastically reduced the total GSH levels. Overall, the exposure to these drugs leads to a significant depletion of the intracellular levels of this important antioxidant, thus contributing to the observed cell death.

### 2.10. Effect of the Drugs on Intracellular Adenosine Triphosphate Levels

Intracellular ATP levels were determined, as previously described [32], through a bioluminescence reaction with luciferase, which catalyzes the formation of light from ATP and luciferin.

As observed in Figure 8, 24 h after exposure to the tested drugs, a significant and concentration-dependent reduction in ATP intracellular levels was detected for all tested drugs, except for mescaline. Furthermore, the presence of a 2-methoxybenzyl substituent leads to a significant increase of drug-mediated ATP depletion. For example, the 2C-N, when tested at a 500 μM, significantly reduced the ATP intracellular levels to about 60% of control cells, while 25B-NBOMe, when tested at the same 500 μM concentration, completely depleted intracellular ATP levels. Overall, these results confirm that the 2C phenethylamines and their *N*-benzylphenethylamine (NBOMe) analogues under study, except mescaline, significantly affect mitochondrial function, leading to a remarkable decrease in ATP production.

### 2.11. Hepatotoxic Profile of 2C Phenethylamines and Their NBOMe Counterparts

The liver is the main organ responsible for the metabolism of drugs, being a target for drug-induced toxicity and for their metabolic activation/detoxification. Therefore, the hepatoxicity of the tested drugs was evaluated in HepG2 cells, 24 h after exposure, by the neutral red uptake and resazurin reduction assays. Figure 9 illustrates the obtained concentration–response (cell death) curves and in Table 4 are depicted the parameters of the fitted curves, namely, the baseline (bottom), the maximum cell death (top), the Hill Slope and the half-maximum-effect concentration (EC_50_). Table 4 also illustrates the curve *p* value obtained for the overall comparison between the 2C-X and 25X-NBOMe curves. As performed in the evaluation of drug-induced cytotoxicity using the SH-SY5Y cells, the EC_50_ values of the fitted curves were also used for statistical comparison of drug cytotoxicity in HepG2 cells. Identical to what was observed for the neuronal model, apart from mescaline, all drugs caused a significant and concentration-dependent cytotoxicity towards HepG2 cells (Figure 9). Additionally, the NBOMe drugs displayed a higher cytotoxic effect than the corresponding 2C phenethylamines, resulting in a substantial and significant shift of the NBOMe concentration–response curves to the left (Figure 9), thus presenting significantly lower EC_50_ values when compared to the corresponding 2C drugs (Table 4).

Once again, mescaline presented the lowest cytotoxic profile of the set of psychedelics tested, not causing a significant cytotoxicity in the range of concentrations tested and in both cytotoxicity assays used. However, mescaline-NBOMe presented a significantly higher cytotoxic effect, with an EC_50_ value of 425.9 µM in the resazurin reduction assay and 476.2 µM in the neutral red uptake assay (Table 4). In accordance with the results obtained in the neuronal model, 25B-NBOMe was the most cytotoxic compound, presenting the lowest EC_50_ values among all the tested drugs (32.82 µM in the resazurin reduction assay and 34.70 µM in the neutral red incorporation assay) (Table 4). Moreover, in agreement with the results observed in differentiated SH-SY5Y cells, the presence of the NBOMe substituent significantly increased their cytotoxic effect on HepG2 cells (significantly higher EC_50_ values obtained for 2C drugs when compared to their NBOMe counterparts—Table 4).

However, in contrast to the observations made in the neuronal model, the HepG2 model showed that the resazurin reduction assay was more sensitive than the neutral red uptake assay for evaluating drug-induced cytotoxicity. This was demonstrated by the lower EC_50_ values obtained for all drugs in the resazurin reduction assay when compared to the EC_50_ values obtained in the neutral red uptake assay (Table 4). Therefore, the resazurin reduction assay was selected to evaluate cell viability in the subsequent experiments.

### 2.12. Effect of the Drugs on Hepatic Oxidative Stress

The ROS intracellular levels were detected in HepG2 cells using the DCFH-DA probe, 24 h after exposure to the different drugs. As observed in differentiated SH-SY5Y cells, none of the tested drugs produced a significant change in the cells’ ROS intracellular levels (Appendix A). The positive control, *t*-BHP (200 µM, 24 h), produced a significant increase in ROS levels when compared to control cells (0 µM).

### 2.13. Effect of Cytochrome P450 Inhibition on Drug-Induced Hepatotoxicity

Analogously to the studies performed in the neuronal model, the impact of the CYP-mediated metabolism on the cytotoxicity of the tested drugs was also evaluated, in HepG2 cells, 24 h after exposure to the drugs in the presence or absence of different CYP inhibitors: 1000 μM ABT, 10 μM quinidine or 1 μM ketoconazole.

No significant changes in mescaline-induced cytotoxicity in the presence or absence of the different CYP inhibitors (Figure 10) were observed, hence suggesting that, under the experimental conditions, these enzymes do not influence the cytotoxicity of this phenethylamine.

However, in ABT-treated cells, a small but significant decrease in resazurin reduction was observed for mescaline-NBOMe (250 and 500 µM), 2C-B (100 and 150 µM), 25B-NBOMe (25 and 50 µM) and 25N-NBOMe (50 and 100 µM), thus demonstrating an increased drug-induced cytotoxicity upon non-selective CYP inhibition (Figure 10). For mescaline-NBOMe and 25N-NBOMe there were no significant changes in resazurin reduction in the cells treated either with quinidine or with ketoconazole, when compared to the cells incubated with the drugs alone (Figure 10). Thus, the obtained results suggest that, in the present experimental conditions, neither CYP2D6 and CYP3A4 impact the cytotoxicity of drugs mescaline-NBOMe and 25N-NBOMe, with the increased cytotoxicity observed in ABT-treated cells highlighting that these drugs are probably detoxified by other cytochrome P450 isoenzyme(s).

Additionally, and as shown in Figure 10, for the drugs 2C-B (150 µM) and 25B-NBOMe (25 µM) there was a significant decrease in the cells’ metabolic capacity when exposed to the drugs in the presence of ketoconazole, which suggest an increased drug-induced cytotoxicity. Therefore, according to the obtained results, the metabolism of 2C-B and 25B-NBOMe appears to be influenced by CYP3A4, as ketoconazole-mediated CYP3A4-inhibition promoted an augmented cytotoxic effect.

On the other hand, for the compound 2C-N (500 µM, Figure 10), there was a significant increase in the metabolic capacity of cells treated with this drug in the presence of quinidine or ketoconazole, thus demonstrating a decreased drug-induced cytotoxicity upon CYP2D6 or CYP3A4 inhibition. Therefore, the obtained results suggest that 2C-N is potentially bioactivated by the CYP2D6 and CYP3A4 isoforms.

In summary, these results suggest that cytochrome P450-mediated metabolism has an impact on the cytotoxicity of these drugs’ functioning, in the case of mescaline-NBOMe, 2C-B, 25B-NBOME and 25N-NBOMe as a detoxification pathway, and in the case of 2C-N as a bioactivation pathway.

### 2.14. Structure–Property–Cytotoxicity Relationships

By analyzing the data from both metabolic and lysosomal activities measurements (cytotoxicity assays) and the lipophilicity and passive permeability of the drugs under study structure–property–cytotoxicity relationships can be established (Figure 11). A correlation between drug lipophilicity and EC_50_ values (Figure 11A) obtained in cell viability assays in both cell lines was observed (Table 3 and Table 4). Except for 2C-B, all the drugs presented higher cytotoxic effects than mescaline-NBOMe, despite its lower lipophilicity. As 2C-B presented a higher passive permeability than mescaline-NBOMe (Table 1) the data suggest that, in terms of cytotoxic effects, the passive permeability has an important role in the cytotoxicity of this type of drugs. This is reinforced by the fact that an inverse logarithmic correlation between passive permeability and the EC_50_ values was obtained (Figure 11B).

Considering that MAOs are located at the mitochondrial membrane, the observed effect on MAO inhibition can be linked with the permeability data of the drugs (Figure 5). In fact, the inhibition of the MAO led to higher cytotoxic effects when neuronal cells were treated with 2C-B rather than mescaline-NBOMe. The same hypothesis can explain the data observed in the mitochondrial membrane potential (Figure 6) and intracellular levels of ATP assays (Figure 8) in differentiated SH-SY5Y cells.

## 3. Materials and Methods

### 3.1. Chemistry

#### 3.1.1. Reagents and General Conditions

The 2,5-Dimethoxybenzaldehyde, 3,4,5-trimethoxybenzaldehyde, 2-methoxybenzaldehyde, ammonium acetate, nitromethane, lithium aluminum hydride (LiAlH_4_), anhydrous sulphate sodium (Na_2_SO_4_), sodium hydroxide (NaOH), potassium hydroxide (KOH), sodium borohydride (NaBH_4_), triethylamine (Et_3_N), elemental bromine (Br_2_), hydrochloric acid ethereal solution, hydrochloric acid (HCl) and nitric acid (HNO_3_) were purchased from Sigma Aldrich (St. Louis, MO, USA) or Alfa-Aesar (Haverhill, MA, USA). All other reagents and solvents were *pro analysis* grade and were acquired from Carlo Erba Reagents (SDS, France) and Scharlab (Barcelona, Spain) and were used without additional purification.

All reactions were monitored by thin-layer chromatography (TLC), performed on 0.2 mm Merck aluminum sheets precoated with silica gel 60 F254 (Algés, Portugal). TLC plates were visualized under a Vilber Lourmat ultra-violet (UV) light with a wavelength of 254 and/or 365 nm.

The reagents and compounds were weighted in a Kern ABJ-NM/ABS-N scale (Balingen, Germany), the solvents were evaporated in a rotary vacuum evaporator Büchi Rotavapor R-210 (Flawil, Switzerland) and the compounds were dried in a MTI Corporation vacuum oven (Porto, Portugal) Short-path vacuum distillation was performed on a Büchi Glass Oven B-585 (Flawil, Switzerland. Advanced automated flash purification from Biotage Isolera™ Prime (Uppsala, Sweden) was also used in the purification (silica gel Merck 0.040–0.063 mm) of some of the synthesized compounds. Microwave-assisted synthesis was performed in a Biotage^®^ Initiator Microwave Synthesizer (Uppsala, Sweden).

^1^H and ^13^C nuclear magnetic resonance (NMR) data were recorded, at room temperature, on a Bruker Avance III (Ettlingen, Germany)operating at 400 and 101 Mega-Hertz (MHz), respectively. Assignments were also made from distortionless enhancement by polarization transfer (DEPT) experiments. In the ^1^H-NMR spectra, the chemical shift (δ) values are reported in ppm relative to tetramethylsilane (TMS), along with the multiplicity of the signal, the number of protons and coupling constants, expressed in Hertz (Hz). In ^13^C-NMR spectra, the chemical shift (δ) values are expressed in ppm with the DEPT spectrum values appearing underlined. Electron spray mass-spectra (ESI-MS) were carried out on an Orbitrap^TM^ Exploris 120 mass spectrometer (Thermo Fischer Scientific, Bremen, Germany). Electron impact mass spectrometry (EI-MS) was done on a Hewlett-Packard spectrometer 5888A (Santa Clara, CA, USA). The MS data are presented as *m*/*z* (% of the relative intensity of the most important fragments).

#### 3.1.2. Synthesis of 2C and NBOMe Drugs

##### Synthesis of β-Nitrostyrene Derivatives

Synthesis of 2,5-Dimethoxy-*β*-nitrostyrene

The 2,5-Dimethoxybenzaldehyde (4.26 g, 26.0 mmol) and ammonium acetate (0.42 g, 5.5 mmol) were dissolved in nitromethane (15 mL) and the reaction refluxed for 7 h. After cooling to room temperature, the solid formed was washed with water and dried in a vacuum oven overnight. Crystallization from hot isopropylalcohol (IPA) yielded a dark yellow/orange solid. The procedure was adapted with modifications from [21]. Yield: 93%. ^1^H NMR (400 MHz, DMSO-*d_6_*): δ = 3.76 (3H, s, 5-OCH_3_), 3.88 (3H, s, 2-OCH_3_), 7.09–7.15 (2H, m, H3 and H4), 7.42 (1H, d, *J* = 2.4 Hz, H6), 8.19 (2H, s, Hα and Hβ). ^13^C NMR (101 MHz, DMSO-*d_6_*): δ = 56.2 (5-OCH_3_), 56.8 (2-OCH_3_), 113.7 (C6), 115.4 (C4), 119.2 (C1), 120.7 (C3), 134.7 (Cβ), 138.8 (Cα), 153.6 (C2), 153.9 (C5). IE-MS *m*/*z* (%): 209 (M^∙+^, 100), 178 (10), 162 (48), 148 (47), 147 (36), 133 (44), 105 (17), 98 (16), 91 (19), 84 (14), 77 (32).

Synthesis of 3,4,5-dimethoxy-*β*-nitrostyrene

A mixture of 3,4,5-trimethoxybenzaldehyde (4.01 g, 20.3 mmol) and ammonium acetate (0.20 g, 2.5 mmol) dissolved in nitromethane (10 mL) was heated under microwave radiation at 150 °C for 6 min. After cooling, the nitromethane was evaporated, and the remaining oil dissolved in diethyl ether and washed with water. The organic layer was dried over anhydrous sulphate sodium (Na_2_SO_4_), filtered and evaporated. The final product was recrystallized from cold water as an orange powder. The procedure was adapted with modifications from [21]. Yield: 73% ^1^H NMR (400 MHz, DMSO-*d_6_*): δ = 3.73 (3H, s, 4-OCH_3_), 3.83 (6H, s, 3-OCH_3_ and 5-OCH_3_), 7.25 (2H, s, H2 and H6), 8.06 (1H, d, *J* = 13.5 Hz, Hα), 8.28 (1H, d, *J* = 13.5 Hz, Hβ). ^13^C NMR (101 MHz, DMSO-*d_6_*): δ = 56.5 (4-OCH_3_), 56.7 (3-OCH_3_), 57.0 (2-OCH_3_), 97.8 (C5), 109.9 (C1), 113.4 (C6), 135.1 (Cβ), 135.7 (Cα), 143.6 (C3), 154.9 (C2), 156.0 (C4). IE-MS *m*/*z* (%): 239 (M^∙+^, 100), 224 (18), 192 (66), 178 (23), 177 (27), 163 (24), 149 (25), 135 (19), 107 (17), 92 (16), 77 (18).

Synthesis of 2,5-Dimethoxyphenethylamine (2C-H) and Mescaline

**General procedure**: A solution of the appropriate *β*-nitrostyrene (8.0 mmol) in anhydrous tetrahydrofuran (30 mL) was added dropwise, under argon, to a stirred suspension of lithium aluminum hydride (LiAlH_4_) (20.0 mmol) in anhydrous tetrahydrofuran (50 mL). The reaction mixture was heated to reflux and stirred for 16 to 36 h. After cooling to room temperature, the excess LiAlH_4_ was destroyed with a small portion of ice. After filtration and removal of the solvent, the residue was diluted in diethyl ether and extracted with 1 M HCl solution. The acidic extract was alkalinized with a 1M NaOH solution and extracted with diethyl ether. The organic layer was dried over anhydrous Na_2_SO_4_, filtered and evaporated. The obtained oil was distilled at 100–120 °C at 0.260 Torr. The salt was obtained by dissolving the residue in IPA and adding 2 M ethereal HCl. The final product was recrystallized from diethyl ether. The procedure was adapted from [21] with modifications.

**2,5-Dimethoxyphenethylamine hydrochloride (2C-H)** Yield: 78%; ^1^H NMR (400 MHz, DMSO-*d_6_*) δ = 2.79–2.87 (2H, m, Hβ), 2.91–2.99 (2H, m, Hα), 3.70 (3H, s, 5-OCH_3_), 3.75 (3H, s, 2-OCH_3_), 6.77–6.82 (2H, m, H4 and H6), 6.91 (1H, d, *J* = 9.1 Hz, H3), 7.98 (3H, *br*s, NH_3_^+^); ^13^C NMR (101 MHz, DMSO-*d_6_*) δ = 28.1 (Cβ), 38.5 (Cα), 55.2 (5-OCH_3_), 55.7 (2-OCH_3_), 111.7 (C3), 112.1 (C4), 116.3 (C6), 126.1 (C1), 151.2 (C2), 153.0 (C5) MS (IE) *m*/*z* (%): 181 (M^∙+^, 36), 152 (100), 137 (62), 121 (36), 108 (10), 91 (20), 77 (24).

**3,4,5-Trimethoxyphenethylamine hydrochloride (Mescaline)** Yield: 42%; ^1^H NMR (400 MHz, DMSO-*d_6_*) δ = 2.80–2.89 (2H, m, Hβ), 2.98–3.08 (2H, m, Hα), 3.03 (3H, s, 4-OCH_3_), 3.77 (6H, s, 3-OCH_3_ and 5-OCH_3_), 6.58 (2H, s, H2 and H6), 8.18 (3H, brs, NH_3_^+^); ^13^C NMR (101 MHz, DMSO-*d_6_*) δ = 33.7 (Cβ), 40.3 (Cα), 56.3 (3-OCH_3_ and 5-OCH_3_), 60.4 (4-OCH_3_), 106.5 (C2 and C6), 133.5 (C1), 136.7 (C4), 153.4 (C3 and C5). IE-MS *m*/*z* (%): 211 (M^∙+^, 48), 194 (18), 182 (100), 181 (71), 167 (69), 151 (25), 136 (18), 121 (10), 95 (10), 80 (18), 77 (12).

Synthesis of 4-Bromo-2,5-dimethoxy-β-phenethylamine (2C-B)

To a solution of 2,5-dimethoxy-*β*-phenethylamine (0.646 g, 3.0 mmol) in acetic acid (15 mL) elemental bromine (Br_2_, 400 μL, 7.8 mmol) was added. After an insoluble dark yellow solid was formed, a 4 M KOH solution was added until basic pH, and the solution extracted with dichloromethane. The organic layer was dried over anhydrous Na_2_SO_4_, filtered and the solvent removed. The dark brown oil was distilled between 130 and 160 °C at 0.215 Torr yielding 153 mg of a pale-yellow residue. The salt was obtained after addition of 2 M HCl ethereal solution and recrystallized from diethyl ether. The procedure was adapted with some modifications from [13]. Yield: 17%; ^1^H NMR (400 MHz, DMSO-*d_6_*) δ = 2.84 (2H, t, *J* = 7.5 Hz, Hβ), 2.98 (2H, t, *J* = 7.4 Hz, Hα), 3.77 (3H, s, 2-OCH_3_), 3.80 (3H, s, 5-OCH_3_), 7.01 (1H, s, H6), 7.20 (1H, s, H3), 7.96 (3H, brs, NH_3_^+^); ^13^C NMR (101 MHz, DMSO-*d_6_*) δ = 28.4 (Cβ), 38.8 (Cα), 56.7 (5-OCH_3_), 57.1 (2-OCH_3_), 109.4 (C4), 115.6 (C3), 115.6 (C6), 126.1 (C1), 149.9 (C5), 152.1 (C2). MS (IE) *m*/*z* (%): 261 (M^∙+^+2, 25), 259 (M^∙+^, 26), 232 (97), 230 (100), 217 (23), 215 (24), 201 (13), 199 (13), 180 (10), 152 (10), 134 (13), 121 (13), 105 (18), 92 (13), 91 (17), 77 (35).

Synthesis of 2,5-Dimethoxy-4-nitro-dimethoxy-β-phenethylamine (2C-N)

Nitric acid (1.8 mL) was added to a solution of 2,5-dimethoxy-*β*-phenethylamine (473 mg, 4.6 mmol) in acetic acid (10 mL). After a solid was formed, a solution of NaOH was added until basic pH and the solution was extracted with dichloromethane. The organic layer was dried over anhydrous Na_2_SO_4_, filtered and the solvent evaporated. The dark brown residue was distilled at 130–160 °C at 0.215 Torr yielding 153 mg of a yellow oil. The salt was obtained after addition of 2 M HCl ethereal solution and recrystallized diethyl ether. The procedure was adapted with modifications from [13]. Yield: 50%; ^1^H NMR (400 MHz, DMSO-*d_6_*) δ: 2.91–2.98 (2H, m, Hβ), 3.00–3,09 (2H, m, Hα), 3.83 (3H, s, 2-OCH_3_), 3.90 (3H, s, 5-OCH_3_), 7.27 (1H, s, H3), 7.51 (1H, s, H6), 7.96 (3H, brs, NH_3_^+^); ^13^C NMR (101 MHz, DMSO-*d_6_*) δ: 28.5 (Cβ), 38.5 (Cα), 56.8 (5-OCH_3_), 57.5 (2-OCH_3_), 107.8 (C3), 117.3 (C6), 133.0 (C1), 138.2 (C4), 146.6 (C5), 151.0 (C2). MS (IE) *m*/*z* (%): 227 (M^∙+^, 1), 226 (7), 197 (100), 182 (3), 167 (26), 137 (6), 136 (2), 120 (20).

Synthesis of N-(2-methoxybenzyl)phenethylamines (NBOMes)

**General procedure**: To a mixture of the corresponding *β*-phenethylamine (1.0 mmol) and the 2-methoxybenzaldehyde (1.1 mmol) in ethanol (10 mL), triethylamine (10 mL) was added. After imine formation, sodium borohydride (2 mmol) was added with agitation for another hour. After solvent evaporation, the residue was dissolved in ethyl acetate (25 mL) and washed with water. The organic layer was dried over Na_2_SO_4_, filtered and evaporated, yielding an oil. The product was purified by flash column chromatography (silica, dichloromethane/methanol/triethylamine (90:9:1). The salts were obtained treating the final oils with 2 M HCl ethereal solution and recrystallized diethyl ether. The procedure was adapted from [22].

**3,4,5-Trimethoxy-*N-*(2-methoxybenzyl)-*β*-phenethylamine (Mescaline-NBOMe)** Yield: 33%; ^1^H NMR (400 MHz, CDCl_3_-*d*_1_) δ: 3.01–3.14 (4H, m, Hα and Hβ), 3.76 (3H, s, 2′-OCH_3_), 3.80 (6H, s, 3-OCH_3_ and 5-OCH_3_), 3.81 (3H, s, 4-OCH_3_), 4.15 (2H, t, *J =* 4.9 Hz, Hγ), 6.37 (2H, s, H2 and H6), 6.84 (1H, dd, *J* = 8.3, 1.0 Hz, H3′), 6.94 (1H, ddd, *J* = 7.4, 7.4, 0.8 Hz, H5′), 7.31 (1H, ddd, *J* = 8.2, 7.6, 1.5 Hz, H4′), 7.41 (1H, dd, *J* = 7.5, 1.5 Hz, H6′), 9.54 (2H, s, NH_2_^+^); 13C NMR (101 MHz, CDCl_3_-*d_1_*) δ: 32.5 (Cα), 46.8 (Cγ), 47.4 (Cβ), 55.5 (2′-OCH_3_) 56.2 (3-OCH_3_ and 5-OCH_3_), 60.8 (4-OCH_3_), 105.7 (C2 and C6), 110.5 (C3′), 118.2 (C1′), 121.1 (C5′), 131.3 (C4′), 132.1 (C6′), 132.1 (C1), 136.9 (C4), 153.5 (C3 and C5), 157.8 (C2′). ESI-MS *m*/*z* (%): 332 (M^+^+1, 100).

**4-Bromo-2,5-dimethoxy-*N-*(2-methoxybenzyl)-*β*-phenethylamine (25B-NBOMe)** Yield: 13%; ^1^H NMR (400 MHz, DMSO *d_6_*) δ: 2.94–3.02 (2H, m, Hβ), 3.03–3.11 (2H, m, Hα), 3.75 (3H, s, 2-OCH_3_), 3.80 (3H, s, 2′-OCH_3_), 3.83 (3H, s, 5-OCH_3_), 4.12 (2H, t, *J* = 5.5 Hz, Hγ), 6.97–7.04 (2H, m, H6 and H3′), 7.06–7.11 (1H, m, H4′), 7.20 (1H, s, H3), 7.38–7.44 (1H, m, H5′), 7.48–7.52 (1H, m, H6′), 9.22 (2H, s, NH_3_^+^); ^13^C NMR (101 MHz, DMSO-*d_6_*) δ: 26.8 (Cα), 45.3 (Cγ), 46.2 (Cβ), 56.1 (2-OCH_3_), 56.7 (5-OCH_3_), 57.1 (2′-OCH_3_), 109.4 (C4), 111.6 (C3′), 115.5 (C6), 116.4 (C3), 120.2 (C1′), 120.8 (C4′), 126.0 (C1), 131.2 (C5′), 131.9 (C6′), 149.9 (C2), 152.0 (C5), 158.0 (C2′). ESI-MS *m*/*z* (%): 380 (M^+^, 97), 381 (M^+^+1, 20), 382 (M^+^+2, 100).

**4-Nitro-2-dimethoxy-*N-*(2-methoxybenzyl)-*β*-phenethylamine (25N-NBOMe)** Yield: 28%; ^1^H NMR (400 MHz, DMSO*-d_6_*) δ: 3.04–3.10 (2H, m, Hβ), 3.11–3.20 (2H, m, Hα), 3.81 (3H, s, 2-OCH_3_), 3.84 (3H, s, 2′-OCH_3_), 3.89 (3H, s, 5′-OCH_3_), 4.14 (2H, t, *J* = 5.4 Hz, Hγ), 7.01 (1H, ddd, *J* = 7.4, 7.4, 1.0 Hz, H5′), 7.09 (1H, dd, *J* = 8.4, 1.0 Hz, H3′), 7.30 (1H, s, H6), 7.39–7.45 (1H, m, H5′), 7.47–7.52 (2H, m, H4 and H6′), 9.21 (2H, m, NH_2_^+^); ^13^C NMR (101 MHz, DMSO*-d_6_*) δ: 26.9 (Cα), 45.4 (Cγ), 45.9 (Cβ), 56.1 (2-OCH_3_), 56.8 (2′-OCH_3_), 57.5 (5-OCH_3_), 107.8 (C3), 111.6 (C3′), 117.2 (C6), 120.1 (C1), 120.9 (C4′), 131.3 (C5′), 131.9 (C6′), 133.0 (C1), 138.2 (C4), 146.7 (C5), 150.9 (C2), 158.0 (C2′). ESI-MS *m*/*z* (%): 347 (M^+^+1, 100).

### 3.2. Evaluation of Drug-like Properties

#### 3.2.1. Reagents and General Conditions

The organic solvents and compounds used to obtain the calibration lines for the analytical determinations were obtained from Carlo Erba, Sigma-Aldrich, TCI or Fluorochem and were HPLC quality grade. The analytical experiments were conducted on a Shimadzu Prominence High-Performance Liquid Chromatograph SPD-M20A system (Shimadzu, Kyoto, Japan) equipped with an autosampler, and a diode array detector (DAD). The HPLC chromatograms were collected using a large bandwidth (190–800 nm) and the retention times of each compound were extracted from the corresponding chromatogram using the Lab Solutions software provided by the manufacturer.

#### 3.2.2. Evaluation of the CHI

The CHI values at pH 7.4 were determined using an experimental protocol described elsewhere [23,25]. The CHI values were assessed from experimental retention times (t_R_) of the compounds under study and a mixture of reference compounds obtained on the HPLC system with a Luna C18 (2) column (150 × 4.6 mm, 5 µm, Phenomenex, CA, USA). Stock solution of compounds in DMSO (10 mM) were diluted in a acetonitrile:water (1:1) mixture to obtain a final concentration of 250 μM. Mobile phase A was 10 mM ammonium acetate (pH 7.4) and mobile phase B was acetonitrile. The following gradient program was applied: 0–7 min 0–100% B, 7–15 min 100% B, 15–17 min 100–0% B. The flow was 1 mL/min, and the injection volume was 40 μL. A calibration curve was obtained using a mixture of the following reference compounds: theophylline, paracetamol, caffeine, benzimidazole, colchicine, carbamazepine, indole, propiophenone, butyrophenone, valerophenone and heptanophenone (Appendix A).

The CHI Log D at pH 7.4 (CHILogD_7.4_) values were back calculated from the experimental CHI values using Equation (1), described elsewhere [23].
CHILogD_7.4_ = 0.0525 CHI − 1.467(1)

#### 3.2.3. Evaluation of the CHI on IAM

The CHI on IAM (CHI (IAM)) values were determined from the experimental t_R_ obtained on the HPLC system with an IAM.PC.DD2 column (100 × 4.6 mm, 10 µm, Regis Technologies, Inc., Morton Grove, IL, USA). Stock solution of compounds in DMSO (10 mM) were diluted in a mixture of water: acetonitrile (1:1) to obtain a final concentration of 250 μM. Mobile phase A was 30 mM ammonium acetate (pH 7.4) and mobile phase B was acetonitrile. The following gradient program was applied: 0–7 min 0–100% B, 7–11 min 100% B, 11–12 min 100–0% B. The flow was 1 mL/min, and the injection volume was 40 μL. A calibration line was obtained using a mixture of the following reference compounds; acetanilide, acetophenone, propiophenone, butyrophenone, valerophenone, hexanophenone, heptanophenone, octanophenone (Appendix A). The Log K_pcell_ (partition between internal and external cellular media) values were estimated from the drugs’ experimental CHI IAM values using Equations (2) and (3) described elsewhere [25].
Log *k*_IAM_ = 0.046 CHI(IAM) + 0.42(2)
Log K_pcell_ = 1.1 Log *k*_IAM_(3)

#### 3.2.4. Evaluation of BBB Permeability

The in vitro BBB permeability of the drugs under study was assessed using the PAMPA-BBB assay, as previously described [33]. The experiments were conducted according to the PiON Inc. (Billerica, MA, USA) instructions and validated through the extrapolation of experimental permeability (log Pe(exp.)) vs. values reported in the literature (log Pe (lit.)) for commercial drugs (verapamil, quinidine, propranolol, lidocaine, progesterone, corticosterone and theophylline) (Appendix A). The PiON inc database software was used for data analysis.

### 3.3. Evaluation of Human Monoamine Oxidase (hMAO) Inhibitory Activity

#### 3.3.1. Materials

Microsomal MAO isoforms prepared from insect cells (BTI-TN-5B1-4) infected with recombinant baculovirus containing cDNA inserts for *h*MAO-A or *h*MAO-B, kynuramine, rasagiline and clorgyline were purchased from Sigma Aldrich.

#### 3.3.2. Human Monoamine Oxidase (*h*MAO) Inhibitory Activity Assay

The *h*MAO inhibitory activities were assessed in microsomal MAO isoforms, prepared from insect cells (BTI-TN-5B1-4) infected with recombinant baculovirus containing cDNA inserts for *h*MAO-A or *h*MAO-B, by measuring the enzymatic conversion rates of kynuramine into 4-hydroxyquinoline. The appropriate amounts of *h*MAO-A and *h*MAO-B were adjusted to obtain, at our experimental conditions, the same maximum velocity (*V*_max_ = 50 pmol·min^−1^) for both isoforms (*h*MAO-A: 3 ng·µL^−1^; *h*MAO-B: 12 ng·µL^−1^). All assays were performed in sodium phosphate buffer solution 50 mM pH 7.4.

The drugs under study or reference inhibitors were pre-incubated at 37 °C for 10 min in the presence of kynuramine (*K*_m_ *h*MAO-A = 20 µM; *K*_m_ *h*MAO-B = 20 µM; final concentration: 2*K*_m_) in 96-well microplates (BRANDplates, pureGrade^TM^, BRAND GMBH, Wertheim, Germany). Then, the reaction was started with the addition of *h*MAO-A or *h*MAO-B. Initial velocities were determined spectrophotometrically in a microplate reader (BioTek Synergy HT from BioTek Instruments, Winooski, VT, USA) at 37 °C by measuring the formation of 4-hydroxyquinoline at 316 nm, over a period of at least 30 min (interval of 1 min). The initial velocities, obtained from the linear phase of product formation, were normalized compared to control, providing the percentages of inhibition of compounds at 10 µM. The percentages of MAO inhibition of each compound were determined from three independent experiments performed in triplicate.

### 3.4. In Vitro Toxicological Studies

#### 3.4.1. Reagents and General Conditions

Dulbecco’s Modified Eagle’s Medium (DMEM)—high glucose (4.5 g/L), sodium bicarbonate, retinoic acid, phorbol 12-myristate 13-acetate (TPA), neutral red solution, resazurin sodium salt, 1-aminobenzotriazole (ABT), quinidine hydrochloride monohydrate, ketoconazole, clorgyline (*N*-methyl-*N*-propargyl-3-(2,4-dichlorophenoxy)propylamine hydrochloride), rasagiline mesylate (*N*-propargyl-1(R)-aminoindan methanesulfonate), *tert*-butyl hydroperoxide solution (Luperox^®^ TBH70X), 2′,7′-dichlorofluorescin diacetate (DCFH-DA), 5,5′-dithiobis (2-nitrobenzoic acid) (DTNB), glutathione reductase from baker’s yeast (*S. cerevisiae*), L-glutathione reduced (GSH), luciferase from *Photinus pyralis* (firefly), bovine serum albumin (BSA), adenosine 5′-triphosphate (ATP) disodium salt hydrate, perchloric acid 70% (HClO_4_), sodium hydroxide (NaOH), potassium bicarbonate (KHCO_3_), di-sodium hydrogen phosphate, sodium dihydrogen phosphate dihydrate, glycine sodium salt hydrate, magnesium sulfate heptahydrate (MgSO_4_x7H_2_O), trizma^®^ base and disodium salt dihydrate (EDTA) were obtained from Sigma-Aldrich (Darmstadt, Germany). Trypsin-EDTA solution (0.25%), heat-inactivated fetal bovine serum (FBS), MEM non-essential amino acids solution and Hanks’ balanced salt solution (HBSS) with or without calcium and magnesium were obtained from Gibco (Winsford, United Kingdom). β-nicotinamide adenine dinucleotide 2′-phosphate reduced tetrasodium salt (NADPH) was obtained from AppliChem (Darmstadt, Germany). DC™ protein assay kit was obtained from Bio-Rad (Hercules, CA, USA). The 2-[2-(3-chlorophenyl)hydrazinylyidene]propanedinitrile (CCCP), mitochondrial membrane potential dye (JC-1, 5,5′,6,6′-tetrachloro-1,1′,3,3′-tetraethylbenzimidazolylcarbocyanine iodide) and D-luciferin potassium salt were obtained from Abcam (United Kingdom). Antibiotic (10,000 U/mL penicillin, 10,000 μg/mL streptomycin) was obtained from Biochrom (Berlin, Germany). Dimethyl sulfoxide (DMSO) was obtained from Merck (Darmstadt, Germany). All sterile plastic material was obtained from Corning Costar (Corning, NY, USA). All the reagents used were of analytical grade or of the highest grade available.

Phenethylamines stock solutions were prepared in water, stored at −20 °C and diluted on the experiment day in cell culture medium.

#### 3.4.2. SH-SY5Y Cell Culture and Differentiation

SH-SY5Y cells were acquired from the American Type Culture Collection (Manassas, VA, USA). Cells were cultured in 25 cm^2^ flasks using DMEM—high glucose supplemented with 10% heat-inactivated FBS, 1% MEM non-essential amino acids and 1% penicillin/streptomycin mixture, and maintained in a 5% CO_2_–95% air atmosphere, at 37 °C, until 80% confluence. Cells were routinely subcultured using trypsin-EDTA and, to avoid phenotypic alterations, were always used between passages 23 and 29. In all experiments, the cells were seeded at a density of 25,000 cells/cm^2^. As previously described, to achieve a dopaminergic neuronal phenotype, the cells were seeded in a complete DMEM medium containing 10 μM retinoic acid and, after 3 days, 80 nM TPA was added to the cultures for another 3 days [26].

#### 3.4.3. HepG2 Cell Culture

HepG2 cells were acquired from the American Type Culture Collection (Manassas, VA, USA). Cells were cultured in 75 cm^2^ flasks using DMEM—high glucose supplemented with a 10% heat inactivated FBS and 1% penicillin/streptomycin mixture, and maintained in a 5% CO_2_–95% air atmosphere, at 37 °C, until 80% confluence. Cells were routinely subcultured using trypsin-EDTA and, to avoid phenotypic alterations, used between passages 16 and 22. In all experiments, the cells were seeded at a density of 250,000 cells/cm^2^ and used 24 h after seeding.

#### 3.4.4. Evaluation of Drugs Cytotoxicity

Neutral red uptake and resazurin reduction assays were performed after exposure of differentiated SH-SY5Y and HepG2 cells to the drugs under study. To obtain a concentration–response curve, a broad concentration range (0–2000 µM) was investigated, and cytotoxicity was evaluated 24 h after exposure in 96-well cell culture plates. As previously described, neutral red uptake assay is based on the incorporation of neutral red dye into the lysosomes of viable cells and the amount of dye incorporated represents the cells’ lysosomal functionality [34], and resazurin reduction assay is based on the irreversible reduction of resazurin dye to the fluorescent resorufin product by metabolically active cells [35].

Briefly, 24 h after being exposed to different concentrations of the compounds, differentiated SH-SY5Y cells were incubated with either neutral red (50 µg/mL) or resazurin (10 µg/mL) solutions for 90 min, at 37 °C, in a 5% CO_2_–95% air atmosphere. For HepG2 cells, 24 h after the exposure to the tested drugs, the cells were incubated with either neutral red (50 µg/mL) or resazurin (10 µg/mL) solutions for 45 and 60 min, respectively, at 37 °C, in a 5% CO_2_–95% air atmosphere.

In the neutral red uptake assay, at the end of the incubation period, the cell culture medium was removed and a solution of absolute ethyl alcohol/distilled water (1:1) with 5% acetic acid was added to extract the dye absorbed only by the viable cells. The absorbance was measured at 540 nm in a multi-well plate reader (PowerWaveX BioTek Instruments, Winooski, VT, USA). In the resazurin assay, at the end of the incubation period, the fluorescence was measured at 530 nm excitation and 590 nm emission wavelengths in a multi-well plate reader (PowerWaveX BioTek Instruments, Winooski, VT, USA).

The results were expressed as a percentage of control cells. At least four independent experiments were performed, in triplicate. For differentiated SH-SY5Y cells, the neutral red uptake assay was identified as the most sensitive assay and, therefore, used for the subsequent experiments. On the other hand, for HepG2 cells, the resazurin reduction assay was identified as the most sensitive assay and, therefore, used for the subsequent experiments.

#### 3.4.5. Assessment of Intracellular Redox State

ROS production was evaluated by using the well-known membrane permeable fluorescent probe 2′,7′-dichlorofluorescein diacetate (DCFH-DA). Both cell lines were seeded in 96-well cell culture plates, and pre-incubated with 10 µM DCFH-DA, protected from light, at 37 °C, in a 5% CO_2_–95% air atmosphere. After 30 min, DCFH-DA was removed, and the cells were exposed to the drugs (0–500 µM for SH-SY5Y cells and 0–1000 µM for HepG2 cells), at 37 °C, in a 5% CO_2_–95% air atmosphere. Subsequently, after 24 h of exposure, the fluorescence was measured at 485 nm excitation and 530 nm emission wavelengths in a multi-well plate reader (PowerWaveX BioTek Instruments, Winooski, VT, USA). Additionally, tert-butyl hydroperoxide (200 µM, 24 h) was used as a positive control for ROS overproduction. The results were expressed as percentage of control cells. At least four independent experiments were performed, in triplicate.

#### 3.4.6. Determination of Cytochrome P450 Inhibition Activity

CYP inhibition assay was used to evaluate the drugs’ potential to inhibit drug-metabolizing enzymes. Accordingly, SH-SY5Y and HepG2 cells were seeded in 96-well cell culture plates and pre-incubated with 1000 µM ABT—a non-isoform specific CYP450 inhibitor, 10 µM quinidine—a CYP2D6 inhibitor or 1 µM ketoconazole—a CYP3A4 inhibitor. After 1 h, the drugs (0–500 µM) were co-incubated with the inhibitors for 24 h, at 37 °C, in a 5% CO_2_–95% air atmosphere. The drugs’ cytotoxicity was then assessed through the neutral red uptake assay for differentiated SH-SY5Y cells, and through the resazurin reduction assay for HepG2 cells, and the results were expressed as percentage of control cells. At least four independent experiments were performed, in triplicate.

#### 3.4.7. Determination of MAO Inhibition Activity

MAO inhibition was used to elucidate the impact of MAO-A and MAO-B in the cytotoxicity of the tested drugs. Briefly, SH-SY5Y cells were seeded and differentiated in 96-well plates and, 6 days after seeding, the cells were pre-incubated with 1 µM clorgyline—MAO-A inhibitor—or 1 µM rasagiline—MAO-B inhibitor. After 1 h, the drugs (0–500 µM) were co-incubated with the inhibitors for 24 h, at 37 °C, in a 5% CO_2_–95% air atmosphere. The drugs’ cytotoxicity was then assessed through the neutral red uptake assay and the results were expressed as percentage of control cells. At least four independent experiments were performed, in triplicate.

#### 3.4.8. Determination of Mitochondrial Membrane Potential

The mitochondrial membrane potential was evaluated by using the JC-1 probe. SH-SY5Y cells were seeded and differentiated in 48-well plates and, 6 days after seeding, the cells were exposed to the six drugs (0–500 µM) for 24 h, at 37 °C, in a 5% CO_2_–95% air atmosphere. Subsequently, the medium was removed, and the cells were incubated with 20 µM JC-1 for 30 min, protected from light, at 37 °C, in a 5% CO_2_–95% air atmosphere. At the end of the incubation period, the cells were centrifuged (300× *g*, 5 min at room temperature) and washed (warm HBSS with calcium and magnesium) twice. The fluorescence was then measured in a multi-well plate reader: at 535 nm excitation and 595 nm emission wavelengths for J-aggregates; and at 485 nm excitation and 535 nm emission wavelengths for JC-1 monomers (PowerWaveX BioTek Instruments, USA). CCCP (100 µM, 4 h) was used as a positive control for mitochondrial depolarization. The results were calculated as red/green fluorescence intensity ratios and expressed as percentages of control cells. At least three independent experiments were performed, in triplicate.

#### 3.4.9. Determination of Intracellular Total Glutathione

Intracellular total glutathione (GSH) was measured through the DTNB-GSH recycling assay. SH-SY5Y cells were seeded and differentiated in 6-well plates and, 6 days after seeding, the differentiated cells were exposed to the six drugs (0–500 µM) for 24 h, at 37 °C, in a 5% CO_2_–95% air atmosphere. Afterwards, the cells were centrifuged (300× *g*, 5 min at 4 °C), the supernatants were rejected, and the cells precipitated with icy 5% HClO_4_, for 30 min, at 4 °C. Then, the suspension was collected, centrifuged (13,000× *g*, 10 min at 4 °C) and the acidic supernatants were stored at −80 °C until a further determination of total GSH.

The acidic samples and standards (GSH standards prepared in 5% HClO_4_) were neutralized with ice-cold 0.76 M KHCO_3_ (1:1) and centrifuged (13,000× *g*, 10 min at 4 °C). Subsequently, 100 µL of the supernatants were transferred to a 96-well plate, followed by the addition of 65 µL of an extemporaneous reagent solution (3.96 mM DTNB and 0.68 mM NADPH prepared in a 7.5 pH phosphate buffer: 71.5 mM Na_2_HPO_4_, 71.5 mM NaH_2_PO_4_, and 0.63 mM EDTA) and the plates were incubated for 15 min in a multi-well plate reader (PowerWaveX BioTek Instruments, Winooski, VT, USA) at 30 °C. At the end of the incubation period, 40 µL of a 10 U/mL glutathione reductase (prepared in phosphate buffer) solution was added and the absorbance was then measured, for 3 min, at 415 nm, in kinetic mode to follow the TNB formation.

The cell pellets were resuspended in 1 M NaOH overnight at 4 °C, and then stored at −20 °C for protein content quantification. Samples protein content was determined using the Bio-Rad DC™ protein assay kit (Bio-Rad Laboratories, USA), according to the manufacturer’s instructions and using BSA as standard. The GSH content in the samples was normalized to the protein content and the results were expressed as a percentage of control cells. At least five independent experiments were performed, in duplicate.

#### 3.4.10. Determination of Intracellular Adenosine Triphosphate

Intracellular adenosine triphosphate (ATP) was quantified through the bioluminescent D-luciferin-luciferase assay. The samples used for the intracellular ATP quantification were prepared as described for the “Intracellular Total Glutathione Quantification” assay. The acidic samples and standards (ATP standards prepared in 5% HClO_4_) were neutralized with ice-cold 0.76 M KHCO_3_ (1:1) and centrifuged (13,000× *g*, 10 min at 4 °C). Afterwards, 75 µL of the supernatants were transferred to a white opaque 96-well plate, followed by the addition of 75 µL of luciferin-luciferase solution (0.15 mM luciferin and 3 × 10^6^ light unit’s luciferase/mL prepared in a 7.6 pH Tris-Glycine buffer solution: 1 mM Trizma; 10 mM MgSO_4_; 50 mM glycine; 0.55 mM EDTA and 1% BSA), protected from light. The samples’ bioluminescence (560 nm) was immediately measured in a multi-well plate reader (PowerWaveX BioTek Instruments, Winooski, VT, USA). The ATP content in the samples was normalized to the protein content (as previously described in “Intracellular Reduced Glutathione Quantification”) and the results were expressed as a percentage of control cells. At least four independent experiments were performed, in duplicate.

### 3.5. Statistical Analysis

GraphPad Prism 8 for Windows (GraphPad Software, San Diego, CA, USA) was used to perform all statistical calculations. The concentration–response curves were drawn using the least squares as the fitting method and the comparisons between curves (LOG EC_50_, TOP, BOTTOM, and Hill Slope) were made using the extra sum-of-squares F test. The normality of the data distribution was evaluated using the KS, D’Agostino and Pearson omnibus and Shapiro-Wilk normality tests. For data with a parametric distribution, one-way ANOVA was used to do the statistical comparisons, followed by Dunnett’s multiple comparisons *post hoc* test (ROS, mitochondrial membrane potential determination, GSH and ATP assays). Statistical comparisons between groups in experiments with two variables were made using two-way ANOVA, followed by the Tukey’s multiple comparison *post hoc* test (CYP and MAO assays). In all cases, *p* values lower than 0.05 were considered significant.

## 4. Conclusions

The use of psychoactive substances is an ancestral practice. If, on the one hand, many of these substances are well studied, on the other hand, many of the NSP lack information regarding their toxicological profile. This collection of data shows that, when compared to their counterparts, NBOMe drugs are significantly more cytotoxic, with much lower EC_50_ values, which can be justified by their higher lipophilicity. For both in vitro models, 25B-NBOMe behaved as the most cytotoxic compound, while mescaline did not cause significant changes in cell viability.

For all the tested drugs, except for mescaline in HepG2 cells, cytochrome P450 seems to be involved in the metabolism of the drugs under study as its inhibition significantly impacted drug-induced cytotoxicity. In the neuronal model, CYP inhibition results in increased drug cytotoxicity for all drugs, therefore suggesting CYP-mediated metabolism as a detoxification pathway. Similarly, in the hepatic model, CYP metabolism seems to function as a detoxification pathway for mescaline-NBOMe, 2C-B, 25B-NBOMe and 25N-NBOMe. On the contrary, for 2C-N CYP inhibition results in decreased drug cytotoxicity, therefore suggesting a CYP-mediated metabolism as a bioactivation pathway. None of the drugs caused a significant change in the ROS intracellular levels.

In differentiated SH-SY5Y cells, all drugs revealed the capacity to induce mitochondrial membrane depolarization, as well as GSH depletion and ATP decline (except for mescaline), with these effects being concentration dependent and more pronounced for the NBOMe derivatives. In agreement with the cytotoxicity and drug-like data, 25B-NBOMe was identified as the most powerful drug in inducing harmful toxic effects in the cells.

## Figures and Tables

**Figure 1 pharmaceuticals-16-01158-f001:**
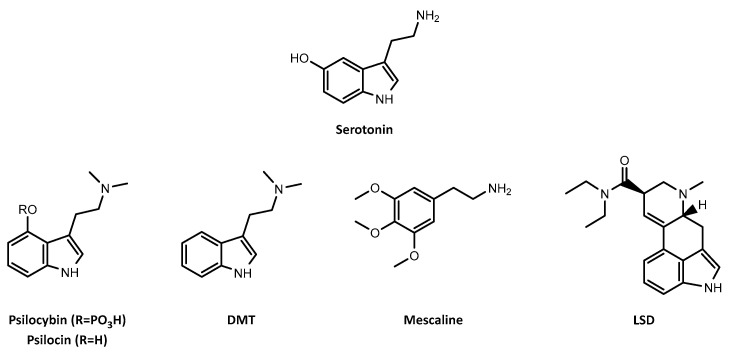
Chemical structures of serotonin and the classic serotonergic psychedelics, psilocybin, and its active metabolite psilocin, *N*,*N*-dimethyltryptamine (DMT), mescaline and lysergic acid diethylamide (LSD).

**Figure 2 pharmaceuticals-16-01158-f002:**
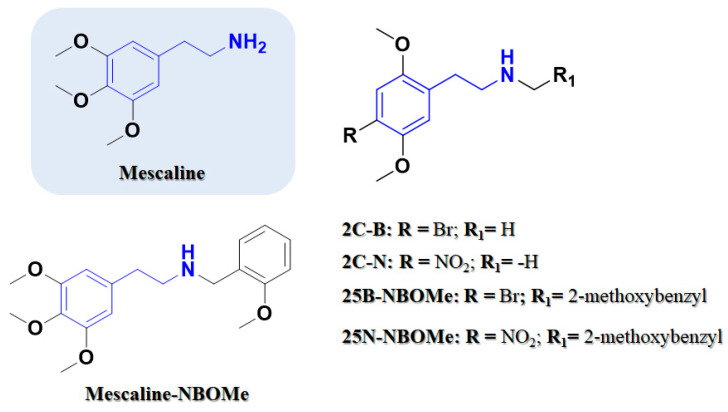
Phenethylamine-based psychedelics. Chemical structures of 2,5-dimethoxyphenethylamines (2C) drugs and their corresponding *N*-(2-methoxybenzyl)phenethylamines (NBOMe) drugs. Phenethylamine scaffold is outlined.

**Figure 3 pharmaceuticals-16-01158-f003:**
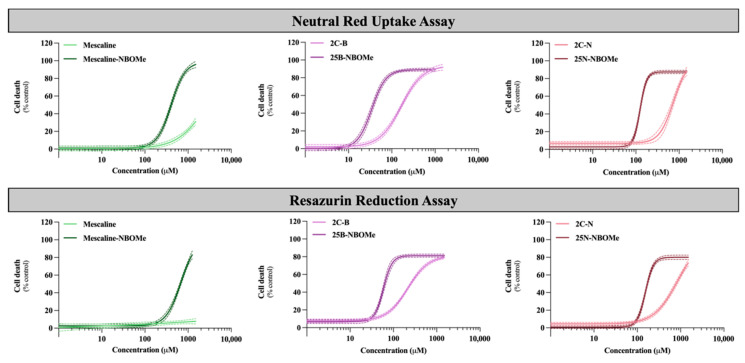
Concentration–response (cell death) curves of the tested drugs (0–1500 µM) obtained, in differentiated SH-SY5Y cells, by the neutral red uptake and the resazurin reduction assays, 24 h after exposure. Results are presented as mean ± 95% CI of at least 4 independent experiments (performed in triplicate). The concentration–response curves were drawn using the least squares method as a fitting method. CI—confidence interval.

**Figure 4 pharmaceuticals-16-01158-f004:**
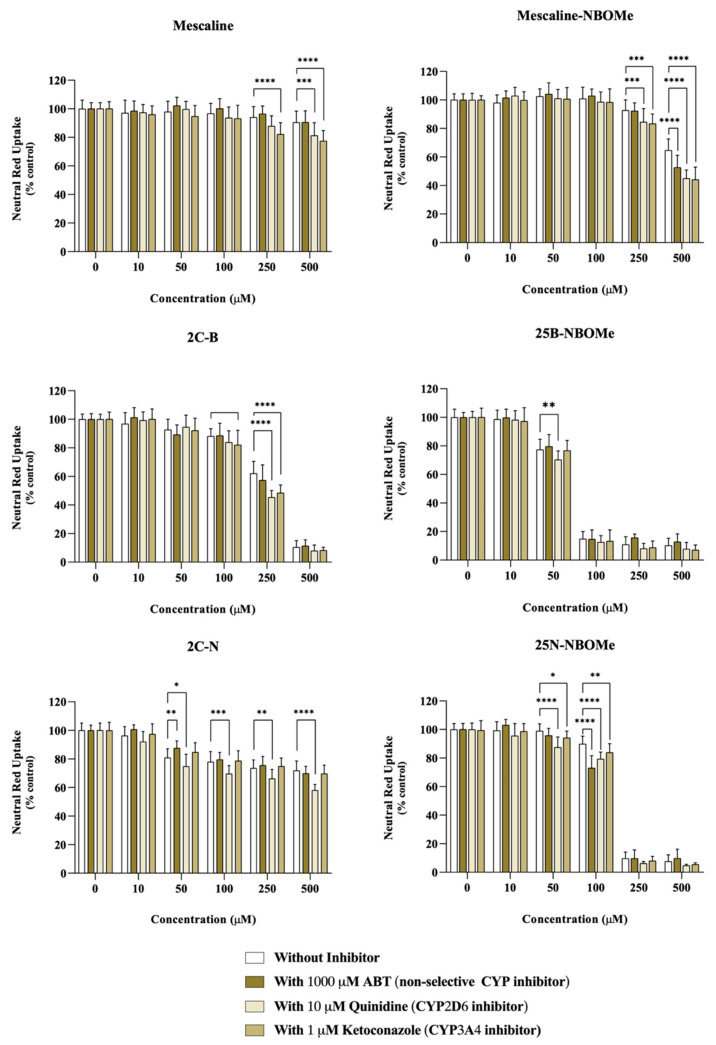
Impact of the cytochrome P450 (CYP)-mediated metabolism on the cytotoxicity of the tested drugs assessed through the neutral red uptake assay, in differentiated SH-SY5Y cells, 24 h after exposure to the drugs in the presence or absence of different CYP inhibitors: 1000 μM ABT (non-selective CYP inhibitor), 10 μM quinidine (CYP2D6 inhibitor) or 1μM ketoconazole (CYP3A4 inhibitor). Results are presented as mean ± SD of at least 4 independent experiments (performed in triplicate). Statistical comparisons were performed using two-way ANOVA, followed by Tukey’s multiple comparison post hoc test [* *p* < 0.05; ** *p* < 0.01; *** *p* < 0.001; **** *p* < 0.0001 vs. control (0 μM)]. In all cases, *p* values < 0.05 were considered statistically significant.

**Figure 5 pharmaceuticals-16-01158-f005:**
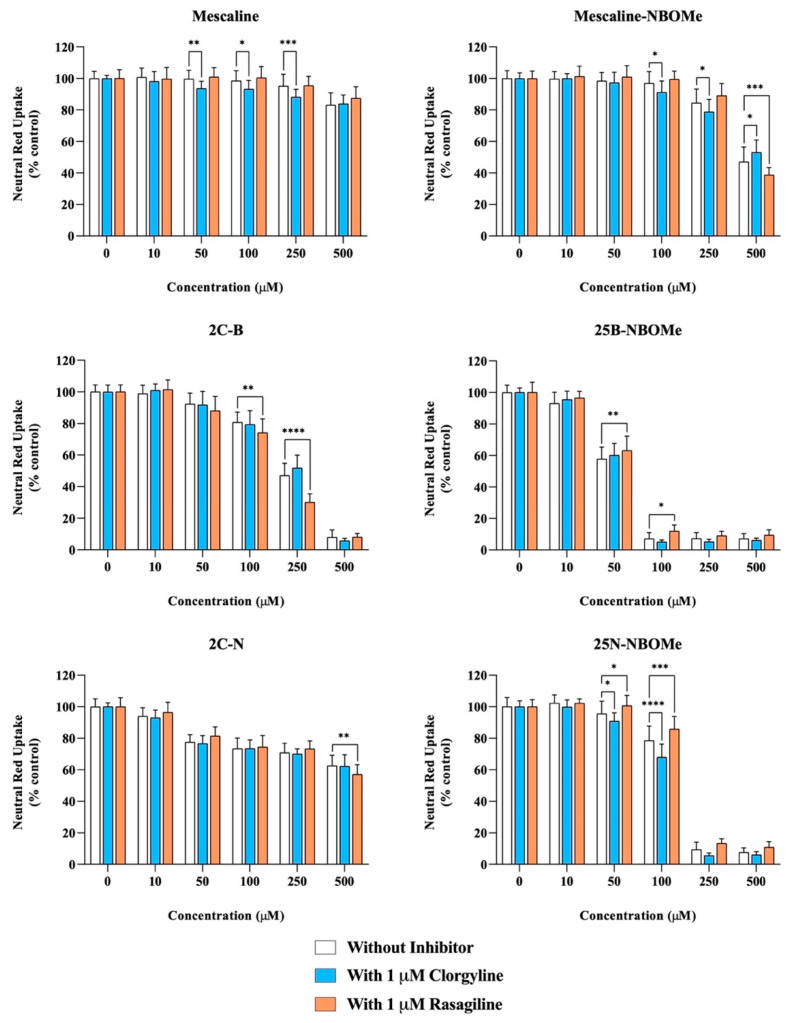
Effect of monoamine oxidase (MAO) inhibition on the drugs-induced cytotoxicity in differentiated SH-SY5Y cells, 24 h after exposure to the tested drugs, in the presence or absence of two MAO inhibitors: 1 μM clorgyline—MAO-A inhibitor or 1 μM rasagiline—MAO-B inhibitor, through the neutral red uptake assay. The results are presented as mean ± SD of at least 4 independent experiments (performed in triplicate). Statistical comparisons were performed using two-way ANOVA, followed by Tukey’s multiple comparison post hoc test [* *p* < 0.05; ** *p* < 0.01; *** *p* < 0.001; **** *p* < 0.0001 vs. control (0 μM)]. In all cases, *p* values < 0.05 were considered statistically significant.

**Figure 6 pharmaceuticals-16-01158-f006:**
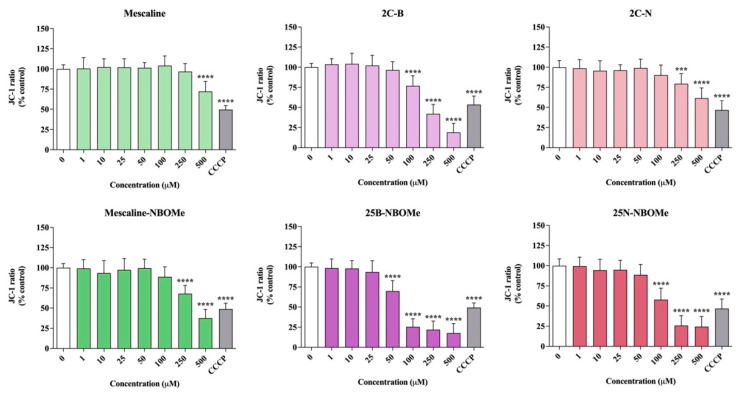
Mitochondrial membrane potential, evaluated with the JC-1 dye, in differentiated SH-SY5Y cells, 24 h after exposure to the tested drugs. The results were calculated as red/green fluorescence intensity ratios and expressed as percentage of control cells. Results are presented as mean ± SD of at least 3 independent experiments (performed in triplicate). As positive control, carbonyl cyanide m-chlorophenyl hydrazone (100 µM, 4 h) was used. Statistical comparisons were performed using one-way ANOVA, followed by Dunnett’s multiple comparison post hoc test. [*** *p* < 0.001; **** *p* < 0.0001 vs. control (0 μM)]. In all cases, *p* values < 0.05 were considered statistically significant.

**Figure 7 pharmaceuticals-16-01158-f007:**
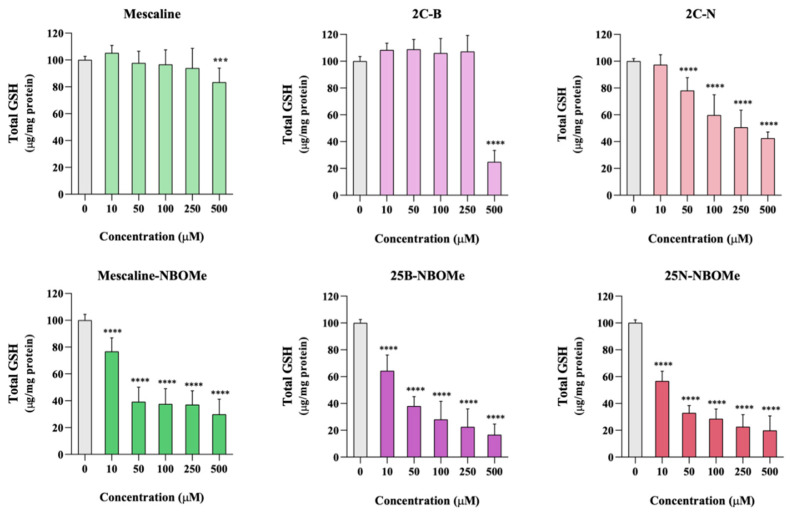
Intracellular levels of total glutathione, evaluated through the DTNB-GSH recycling assay, in differentiated SH-SY5Y cells, 24 h after exposure to the tested drugs. Results are presented as mean ± SD of at least 5 independent experiments (performed in duplicate). Statistical comparisons were performed using one-way ANOVA, followed by Dunnett’s multiple comparison post hoc test. [*** *p* < 0.001; **** *p* < 0.0001 vs. control (0 μM)]. In all cases, *p* values < 0.05 were considered statistically significant.

**Figure 8 pharmaceuticals-16-01158-f008:**
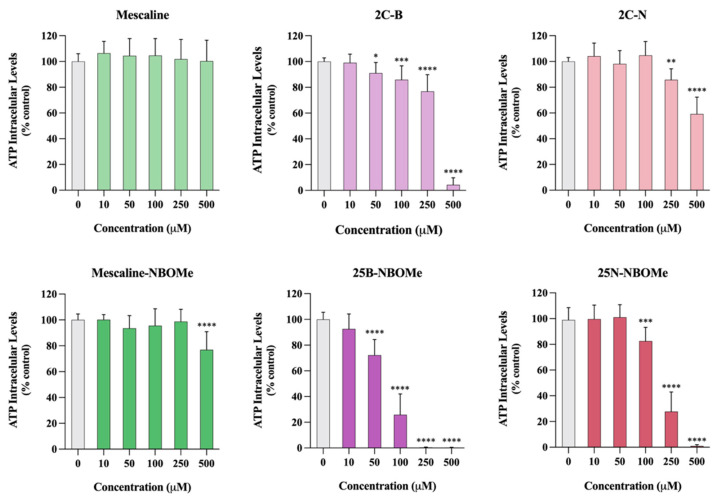
Intracellular adenosine triphosphate (ATP) levels, evaluated through an ATP bioluminescence assay, in differentiated SH-SY5Y cells, 24 h after exposure to the tested drugs. Results are presented as mean ± SD of at least 4 independent experiments (performed in duplicate). Statistical comparisons were performed using one-way ANOVA, followed by Dunnett’s multiple comparison post hoc test. [* *p* < 0.05; ** *p* < 0.01; *** *p* < 0.001; **** *p* < 0.0001 vs. control (0 μM)]. In all cases, *p* values < 0.05 were considered statistically significant.

**Figure 9 pharmaceuticals-16-01158-f009:**
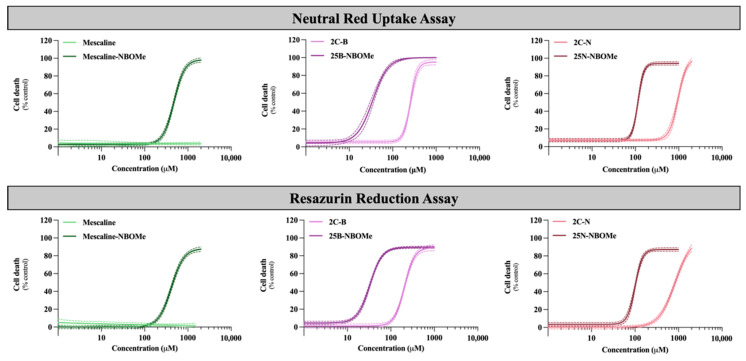
Concentration–response (cell death) curves of the tested drugs (0–2000 µM) obtained in HepG2 cells by the neutral red uptake and the resazurin reduction assays, 24 h after exposure. Results are presented as mean ± 95% CI of at least 4 independent experiments (performed in triplicate). The concentration–response curves were drawn using the least squares method as a fitting method. CI—confidence interval.

**Figure 10 pharmaceuticals-16-01158-f010:**
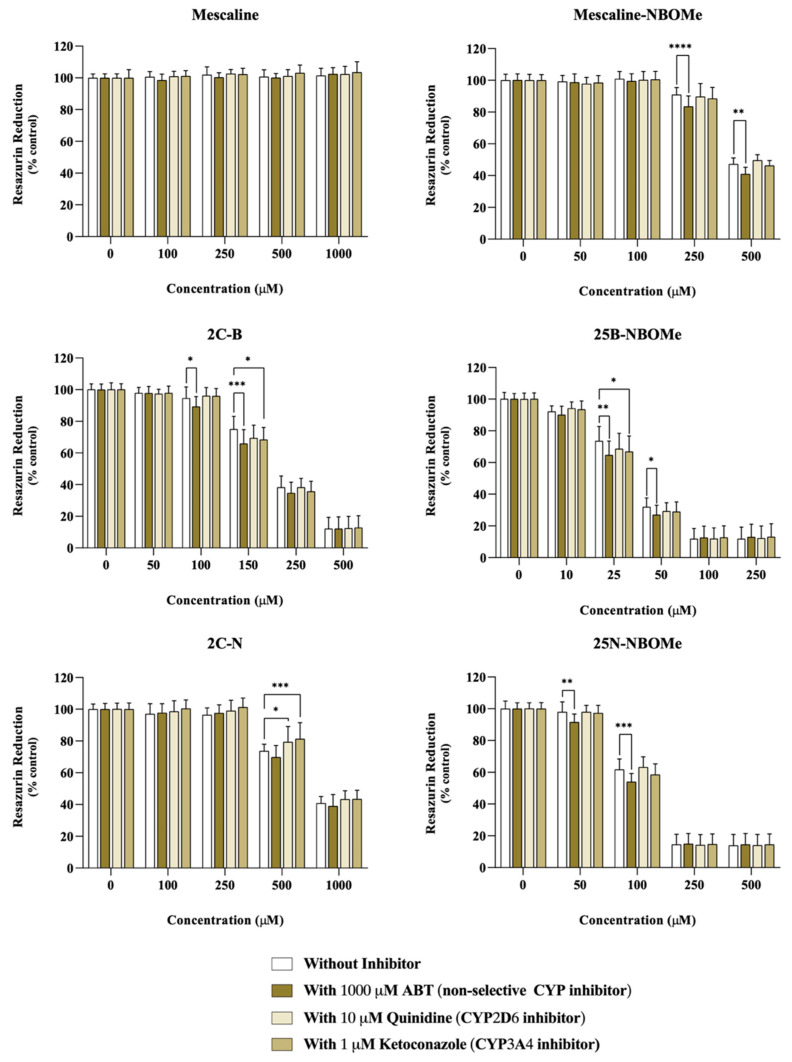
Impact of the metabolism via cytochrome P450 (CYP) on the cytotoxicity of the tested drugs assessed through the resazurin reduction uptake assay, in HepG2 cells, 24 h after exposure to the drugs in the presence or absence of different CYP inhibitors: 1000 μM ABT (non-selective CYP inhibitor), 10 μM quinidine (CYP2D6 inhibitor) or 1μM ketoconazole (CYP3A4 inhibitor). Results are presented as mean ± SD of at least 4 independent experiments (performed in triplicate). Statistical comparisons were performed using two-way ANOVA, followed by Tukey’s multiple comparison post hoc test [* *p* < 0.05; ** *p* < 0.01; *** *p* < 0.001; **** *p* < 0.0001 vs. control (0 μM)]. In all cases, *p* values < 0.05 were considered statistically significant.

**Figure 11 pharmaceuticals-16-01158-f011:**
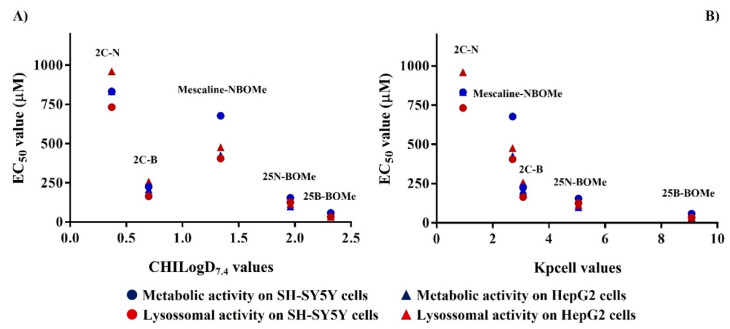
Correlations between EC_50_ values obtained in both metabolic and lysosomal activity measurements (cytotoxicity assays) in both cell lines tested with the lipophilicity (**A**) and calculated passive permeability (**B**).

**Table 1 pharmaceuticals-16-01158-t001:** Chromatographic Hydrophobicity Index (CHILogD_7.4_) and partition between internal and external cellular media (K_pcell_), of 2C-B, 2C-N, mescaline-NBOMe, 25B-NBOMe and 25N-NBOMe drugs.

Drugs	CHI LogD_7.4_	K_pcell_	HBD ^a^	NRB ^a^	TPSA ^a^ (Å^2^)	BBB Prediction ^a^
2C-B	0.70	3.08	1	4	46.1	No
2C-N	0.37	0.94	1	5	91.9	No
Mescaline-NBOMe	1.34	2.71	1	8	53.5	Yes
25B-NBOMe	2.32	9.08	1	8	44.3	Yes
25N-NBOMe	1.96	5.05	1	9	90.1	Yes

^a^ Values obtained from SwissADME; Number of Hydrogen Bond Donors (HBD), Calculated Topological Polar Surface Area (TPSA); Number of Rotatable Bonds (NRB).

**Table 2 pharmaceuticals-16-01158-t002:** Permeability (−Log Pe) of standards and NBOMe drugs in the PAMPA-BBB assay.

Compounds	−Log Pe ± SD	Prediction
Verapamil	4.4 ± 0.34	${\mathrm{CNS}}^{+}$
Lidocaine	4.8 ± 0.39	${\mathrm{CNS}}^{+}$
Quinidine. HCl	4.8 ± 0.39	${\mathrm{CNS}}^{+}$
Progesterone	4.5 ± 0.44	${\mathrm{CNS}}^{+}$
Propanolol. HCl	4.6 ± 0.42	${\mathrm{CNS}}^{+}$
Theophilline	6.8 ± 0.39	${\mathrm{CNS}}^{-}$
Corticosterone	5.0 ± 0.37	${\mathrm{CNS}}^{\pm }$
Mescaline-NBOMe.HCl	4.7 ± 0.49	${\mathrm{CNS}}^{+}$
25B-NBOMe.HCl	4.4 ± 0.36	${\mathrm{CNS}}^{+}$
25N-NBOMe.HCl	4.6 ± 0.34	${\mathrm{CNS}}^{+}$

**Table 3 pharmaceuticals-16-01158-t003:** EC_50_ (half-maximum-effect concentration), TOP (maximal effect), BOTTOM (baseline) and Hill Slope values of the concentration-response curves of the tested drugs (0–1500 µM), obtained in differentiated SH-SY5Y cells by the neutral red uptake and the resazurin reduction assays, 24 h after exposure, obtained. Concentration–response curves were fitted using least squares as the fitting method and the comparisons between the curves were made using the extra sum-of-squares F test. [**** *p* < 0.0001 2C-X vs. 25X-NBOMe]. Mesc—Mescaline; NA—Not applicable.

	Neutral Red Uptake					
	Mesc	2C-B	2C-N	Mesc-NBOMe	25B-NBOMe	25N-NBOMe
EC_50_	NA	164.6	732.7	405.6 (****)	33.86 (****)	125.0 (****)
Top	NA	93.68	≈100	98.57	87.85	87.38
Bottom	NA	1.900	6.911	1.218	0.08250	2.841
Hill Slope	NA	1.746	2.691	2.681	2.687	5.758
Curve *p* value (comparison between the 2C-X and 25X-NBOMe curves)	-	-	-	<0.0001	<0.0001	<0.0001
	**Resazurin Reduction**					
EC_50_	NA	224.9	832.0	677.2 (****)	58.36 (****)	154.1 (****)
Top	NA	83.95	≈100	94.72	80.62	80.10
Bottom	NA	7.210	4.440	2.774	6.827	0.9223
Hill Slope	NA	1.670	1.627	2.524	4.464	4.692
Curve *p* value (comparison between the 2C-X and 25X-NBOMe curves)	-	-	-	<0.0001	<0.0001	<0.0001

**Table 4 pharmaceuticals-16-01158-t004:** EC_50_ (half-maximum-effect concentrations), TOP (maximal effect), BOTTOM (baseline) and Hill Slope values of the concentration–response of the tested drugs (0–2000 µM), obtained in HepG2 cells by the neutral red uptake and the resazurin reduction assays, 24 h after exposure, Concentration–response curves were fitted using least squares as the fitting method and the comparisons between *the* curves were made using the extra sum-of-squares F test. [**** *p* < 0.0001 2C-X vs. 2C-X-NBOMe]. Mesc—Mescaline; NA—Not applicable.

	Neutral Red Uptake					
	Mesc	2C-B	2C-N	Mesc-NBOMe	25B-NBOMe	25N-NBOMe
EC_50_	NA	257.2	960.0	476.2 (****)	34.70 (****)	114.7 (****)
Top	NA	95.01	≈100	98.25	≈100	94.13
Bottom	NA	5.433	7.328	2.853	4.279	7.254
Hill Slope	NA	5.413	4.340	3.666	2.442	6.518
Curve *p* value (comparison between the 2C-X and 25X-NBOMe curves)	-	-	-	<0.0001	<0.0001	<0.0001
	**Resazurin Reduction**					
EC_50_	NA	206.0	833.9	425.9 (****)	32.82 (****)	99.68 (****)
Top	NA	89.29	≈100	87.43	89.49	87.26
Bottom	NA	1.072	0.2545	−0.3754	4.552	3.464
Hill Slope	NA	3.731	2.334	3.411	3.079	5.089
Curve *p* value (comparison between the 2C-X and 25X-NBOMe curves)	-	-	-	<0.0001	<0.0001	<0.0001

## Data Availability

Data is contained within the article and Appendix A.

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
