# Peer review of "Unraveling the In Vitro Toxicity Profile of Psychedelic 2C Phenethylamines and Their N-Benzylphenethylamine (NBOMe) Analogues"

_pharmaceuticals, 2023, doi:10.3390/ph16081158_

Round 1

Reviewer 1 Report

The manuscript is of high impact for the medicinal chemistry community. Only some minor typos are present such as (line 38remove the s: …, s cytotoxicity of these drugs….

Moreover, throughout the entire paper an alert sentence (Error! Reference source not found) is present. It is probably originated by the proof preparation process as it seems impossible that the authors didn’t detect it in the manuscript preparation.

It is my opinion that, once these minor issues will be addressed the manuscript could be published with no further revision.

Author Response

The manuscript is of high impact for the medicinal chemistry community. Only some minor typos are present such as (line 38remove the s: …, s cytotoxicity of these drugs….

The typo was corrected

Moreover, throughout the entire paper an alert sentence (Error! Reference source not found) is present. It is probably originated by the proof preparation process as it seems impossible that the authors didn’t detect it in the manuscript preparation.

The software problem was amended.

Reviewer 2 Report

This study presented many assays of Mescaline and its derivatives, including drug lipophilicity assay, drug blood-brain barrier permeability assay, drug cytotoxicity assay, human monoamine oxidase inhibition assay etc. overall, this paper is a superposition of various assays, it is not very pleasant to read through it, but it is still publishable if a minor issue is addressed.

 1.       It seems that the authors’ citation software had problems, “Error! Reference source not found” showed many times, this error should be corrected.

Author Response

  1. It seems that the authors’ citation software had problems, “Error! Reference source not found” showed many times, this error should be corrected.

 The software problem was amended.

Reviewer 3 Report

This manuscript describes the in vitro toxicity evaluation of mescalin and its derivatives, evaluating their neurotoxicity and hepatotoxicity in vitro, and evaluating the main pathways of cytotoxicity from various aspects such as oxidative stress and metabolism.

However, there are still problems and defects in content and format

1. There are some problems in the format. In the second part of Materials and methods, the sub-title is incorrect, skipping 2.2, and there are redundant "-" in the sub-headings of 2.3.1 and 2.3.2. The second title of the third part is out of order. And there are two fourth parts.

2. There is a problem in the reference part of the drawing annotation, and the "Error! Reference source not found"

3. The content of CHI can be enriched appropriately for easy understanding.

4. The contents of Table 4 are not explained in the text. Please describe them in detail. Two of them appear in Figure 11. Please mark the correct notes.

Please check the context especially  about literatures.

Author Response

However, there are still problems and defects in content and format

  1. There are some problems in the format. In the second part of Materials and methods, the sub-title is incorrect, skipping 2.2, and there are redundant "-" in the sub-headings of 2.3.1 and 2.3.2. The second title of the third part is out of order. And there are two fourth parts.

All the corrections were done on the manuscript. Thank you for the valid input.

  1. There is a problem in the reference part of the drawing annotation, and the "Error! Reference source not found"

The software problem was amended.

  1. The content of CHI can be enriched appropriately for easy understanding.

A brief outline was added to the text.

  1. The contents of Table 4 are not explained in the text. Please describe them in detail. Two of them appear in Figure 11. Please mark the correct notes.

We acknowledge the Reviewer suggestion and the contents of Table 3 e and 4 [parameters of the fitted concentration-response (cell death) curves] were now mentioned in the revised version of the manuscript. The overall comparison of the fitted curves (Curve p value for comparison between the 2C-X and 25X-NBOMe curves) was also added to the revised version of Table 3 (studies in SH-SY5Y cells) and Table 4 (studies in HepG2 cells). Accordingly, the following information was added to the revised manuscript:

Figure 3 illustrates the obtained concentration-response (cell death) curves and in Table 3 are depicted the parameters of the fitted curves, namely, the baseline (bottom), the maximum cell death (top), the hill slope and the half-maximum-effect concentration (EC50). In Table 3 is also illustrated the curve p value obtained for the overall comparison between the 2C-X and 25X-NBOMe curves. The EC50 values of the fitted curves were used for statistical comparison of drug-induced cytotoxicity.”

Figure 11 illustrates the obtained concentration-response (cell death) curves and in Table 4 are depicted the parameters of the fitted curves, namely, the baseline (bottom), the maximum cell death (top), the hill slope and the half-maximum-effect concentration (EC50). In Table 4 is also illustrated the curve p value obtained for the overall comparison between the 2C-X and 25X-NBOMe curves. As performed in the evaluation of drug-induced cytotoxicity using the SH-SY5Y cells, the EC50 values of the fitted curves were also used for statistical comparison of drug cytotoxicity in HepG2 cells.”